# Archetypal Analysis for population genetics

**Julia Gimbernat-Mayol[1], Albert Dominguez Mantes[2,3,4], Carlos D. Bustamante[4], Daniel Mas Montserrat[4], Alexander G. Ioannidis[4,5]***

**1** Department of Bioengineering, Faculty of Engineering, Imperial College London, London, United Kingdom, **2** Brain Mind Institute, School of Life Sciences, École Polytechnique Fédérale de Lausanne, Lausanne, Switzerland, **3** Institute of Bioengineering, School of Life Sciences, École Polytechnique Fédérale de Lausanne, Lausanne, Switzerland, **4** Department of Biomedical Data Science, Stanford Medical School, Stanford, California, United States of America, **5** Institute for Computational and Mathematical Engineering, Stanford University, Stanford, California, United States of America

* ioannidis@stanford.edu

**Data Availability Statement:** The code is available at https://github.com/AI-sandbox/archetypal-analysis. All relevant data are within the references of the manuscript and its Supporting information files.

## Abstract

The estimation of genetic clusters using genomic data has application from genome-wide association studies (GWAS) to demographic history to polygenic risk scores (PRS) and is expected to play an important role in the analyses of increasingly diverse, large-scale cohorts. However, existing methods are computationally-intensive, prohibitively so in the case of nationwide biobanks. Here we explore Archetypal Analysis as an efficient, unsupervised approach for identifying genetic clusters and for associating individuals with them. Such unsupervised approaches help avoid conflating socially constructed ethnic labels with genetic clusters by eliminating the need for exogenous training labels. We show that Archetypal Analysis yields similar cluster structure to existing unsupervised methods such as ADMIXTURE and provides interpretative advantages. More importantly, we show that since Archetypal Analysis can be used with lower-dimensional representations of genetic data, significant reductions in computational time and memory requirements are possible. When Archetypal Analysis is run in such a fashion, it takes several orders of magnitude less compute time than the current standard, ADMIXTURE. Finally, we demonstrate uses ranging across datasets from humans to canids.

## Author summary

This work introduces a method that combines the singular value decomposition (SVD) with Archetypal Analysis to perform fast and accurate genetic clustering by first reducing the dimensionality of the space of genomic sequences. Each sequence is described as a convex combination (admixture) of archetypes (cluster representatives) in the reduced dimensional space. We compare this interpretable approach to the widely used genetic clustering algorithm, ADMIXTURE, and show that, without significant degradation in performance, Archetypal Analysis outperforms, offering shorter run times and representational advantages. We include theoretical, qualitative, and quantitative comparisons between both methods.

This is a *PLOS Computational Biology* Methods paper.

**Funding:** This work was supported in part by the Chan Zuckerberg Biohub (awarded to CDB) and by the Royal Academy of Engineering Leaders Scholarship (awarded to JGM). The funders had no role in study design, data collection and analysis, decision to publish, or preparation of the manuscript.

# Introduction

Estimating ancestry cluster allele frequencies and cluster membership from single nucleotide polymorphism (SNP) data is important for many applications in population genetics and applying such methods to characterize diverse human cohorts has become an essential part of large-scale genomic studies. With the growing number of samples in whole genome databases, efficient population clustering techniques that can handle such sample sizes have become increasingly important. Existing techniques for the clustering of genomes include STRUCTURE [1], FRAPPE [2] and, ADMIXTURE [3]. These compute probabilistic values referred to as *ancestry coefficients* that represent the fraction of the genome of an individual attributable to a particular population cluster. These methods can perform both supervised and unsupervised inference of *ancestry coefficients*. Supervised inference requires reference individuals from pre-defined ancestral populations, while unsupervised inference uses the structure of the genome-wide data alone. These existing approaches perform inference via Bayesian [1] or likelihood based methods [2, 3] and tend to be computationally expensive due to the high dimensionality of genomic data.

Dimensionality reduction techniques such as multidimensional scaling (MDS), principal component analysis (PCA) and uniform manifold approximation (UMAP) have been used to overcome the high dimensionality of genomic data [4, 5], and have become indispensable for visualization and representation of diversity amongst genomic sequences. In PCA, samples are projected onto the axes of highest variation, each of which is a linear combination of allelic dosages across variants [6]. This method has become particularly important in genome-wide association studies and has also been used to investigate the distribution of genetic variation across geography [7]. An advantage is that no assumptions are made about ancestral populations; however, interpretation can often be misleading if sampling designs are irregular. Unsupervised clustering techniques such as ADMIXTURE or Archetypal Analysis (AA) can complement PCA to provide a detailed description of data and to augment visualization. In this work we show how AA can be coupled with PCA, specifically the Single Value Decomposition (SVD), to efficiently cluster samples providing shorter run-times than STRUCTURE or ADMIXTURE. We also discuss how these techniques work, where they differ, and how they relate to well established general-purpose clustering techniques such as K-Means and K-Medioids.

# Materials and methods

## System overview

The complete proposed pipeline is presented in Fig 1.

**Singular value decomposition.**   If we observe $N$ individuals at $M$ SNP positions, each individual $i$ can be represented by a genotype vector $\mathbf{x}_i \in \{0, \frac{1}{2}, 1\}^M$, where each position $j$ in $\mathbf{x}_i$ indicates the average number of alternate alleles found for each $j$ (position) and $i$ (individual's diploid genome). By aggregating $\mathbf{x}_i$ the vectors for all individuals, we obtain an $M \times N$ genotype matrix $\mathbf{X} = [\boldsymbol{x_1}, \ldots, \boldsymbol{x_N}]$. We center the columns of $\mathbf{X}$ to produce data matrix $\mathbf{X_c}$ of centered genotype vectors and then compute the SVD:

$$\mathbf{X_c} = \mathbf{U}\Sigma\mathbf{V}^T \tag{1}$$

This yields $\mathbf{U}$ and $\mathbf{V}$, the left and right-singular vectors respectively. The first $N-1$ scores $\mathbf{X}' = \mathbf{U}\Sigma$ can then be used as input for Archetypal Analysis. As described in [6] these vectors are made up of a linear combination (rotation) of genotypic values across the genome.

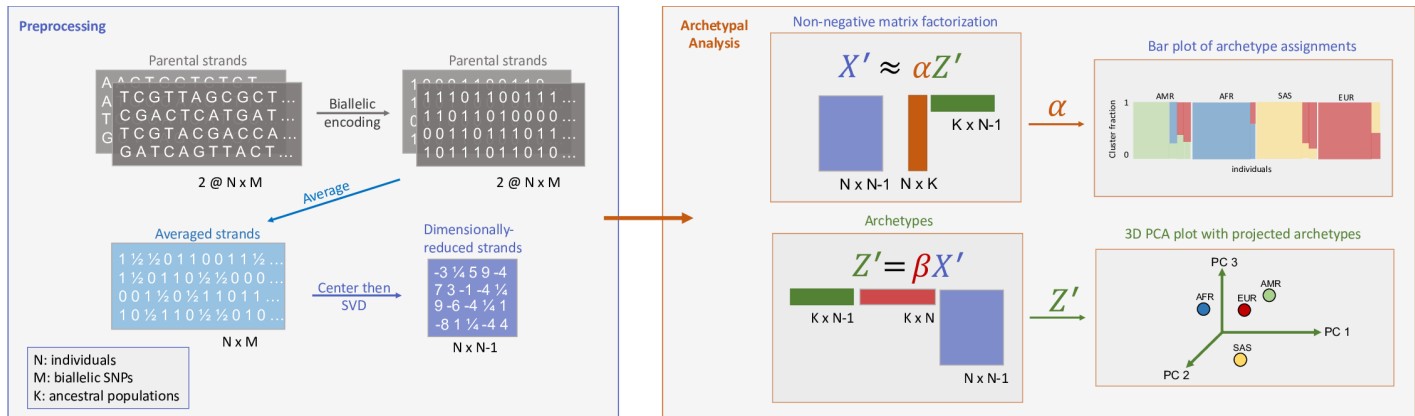

**Fig 1. Archetypal Analysis pipeline.** The allele counts from both haplotypes of each of $N$ individuals are averaged and then dimensionally-reduced from $M$ SNPs to $N - 1$ element singular vectors via the SVD. Archetypal Analysis then implements an alternating non-negative matrix factorization algorithm that minimizes a constrained sum of squares to find ancestry proportions ($\alpha$) and cluster centroids ($Z'$; archetypes, $Z' = ZV^T$). Archetypal analysis models the individual genotypes as originating from the admixture of $K$ parental populations, where $K$ is an input parameter. For visualization we create bar plots for proportions of archetype assignments given by the matrix $\alpha$, and project archetypes $Z$ into a 3D subspace using the first three principal components of the individual genotype sequences.

Because the subspace spanned by the centered genotype vectors can have no more than $N - 1$ dimensions with $N$ the number of samples, there is no loss of information in projecting these centered genotype vectors onto their $N - 1$ right singular vectors before applying Archetypal Analysis. This operation corresponds simply to a rotation of the coordinate system followed by a pruning of the unused dimensions and yields a space that generally has much smaller dimensionality ($N - 1$) than the original space ($M$, number of genotyped positions), since typically $N \ll M$.

**Archetypal Analysis.** This non-negative matrix factorization method was first developed by Cutler and Breiman in 1994 [8], and here it represents each individual as a convex combination of *extreme points*, or archetypes, in allele frequency space. In particular, given an $N$ x $M$ multivariate data set $X$ with $N$ individuals and $M$ SNPs, for a given number of archetypes or clusters $K$, the algorithm finds the $M$ x $K$ matrix of archetypes $Z$ according to two principles:

1. The samples are approximated as convex combinations of the archetypes such that the residual sum of squares (RSS) between the approximation and original data is minimized:

$$RSS = ||X - \alpha Z^T||_F^2 \tag{2}$$

with $|| \cdot ||_F^2$ representing the squared Frobenius norm, $\alpha$ representing the fractional ancestry assignments, so $\sum_{j=1}^{K} \alpha_{ij} = 1$, $1 \geq \alpha_{ij} \geq 0$ for $i = 1, \ldots, N$, and $j = 1, \ldots, K$.

2. The archetypes are convex combinations of the samples:

$$Z = X^T \beta \tag{3}$$

with $\beta$ an $N \times K$ matrix and $\beta_{ij}$ indicating the weight of sample $i$ at archetype $j$, and $\sum_{i=1}^{N} \beta_{ij} = 1$ with $1 \geq \beta_{ij} \geq 0$.

By combining Eqs 2 and 3 we have:

$$RSS = ||X - \alpha \beta^T X||_F^2 = \sum_i ||x_i - \sum_k \alpha_{ik} \sum_j \beta_{kj} x_j||_2^2 \tag{4}$$

Where $|| \cdot ||_2^2$ represents the squared L2 norm.

The optimization problem presented in Eqs 2 and 3 consists of finding the weight matrices $\boldsymbol{\alpha}$ and $\boldsymbol{\beta}$ for a given data matrix $\boldsymbol{X}$ and a particular number of archetypes $K$. This is commonly solved through an iterative process of optimizing $\boldsymbol{\alpha}$ and $\boldsymbol{\beta}$ in an alternating fashion. For a fixed set of values for $\boldsymbol{\alpha}$, finding the optimal values for $\boldsymbol{\beta}$ is reduced to a constrained least squares problems, and vice versa [8]. The iterative process is typically repeated until the quality of the decomposition reaches a pre-defined threshold, or up to a fixed maximum number of steps. The constrained least square optimization problem can be solved through a variety of techniques. Here we make use of the implementation of [9], which utilizes a non-negative least squares solver obtaining $\alpha_{ij} \geq 0$ and $\beta_{ij} \geq 0$, where an extra dimension is added to enforce $\sum_{i=1}^{n} \alpha_{ij} = 1$ and $\sum_{i=1}^{n} \beta_{ij} = 1$. There are multiple open-source packages available in R [10], Python [9] and MATLAB [11] that implement Archetypal Analysis.

Unlike ADMIXTURE, Archetypal Analysis permits the use of rotated and projected (dimensionally reduced) representations of SNP data. If all singular vectors are used, the residual sum of squares of the decomposition ($RSS'$) using projected data $\boldsymbol{X'}$ is equivalent to the $RSS$ of the original decomposition:

$$RSS' = \sum_i ||\boldsymbol{x_i'} - \sum_k \alpha_{ik} \sum_j \beta_{kj} \boldsymbol{x_j'}||_2^2 = \sum_i ||\boldsymbol{P}(\boldsymbol{x_i} - \boldsymbol{\mu}) - \sum_k \alpha_{ik} \sum_j \beta_{kj} \boldsymbol{P}(\boldsymbol{x_i} - \boldsymbol{\mu})||_2^2$$

$$= \sum_i ||\boldsymbol{P}\boldsymbol{x_i} - \boldsymbol{P}(\sum_k \alpha_{ik} \sum_j \beta_{kj} \boldsymbol{x_j})||_2^2 \qquad (5)$$

$$= \sum_i ||\boldsymbol{x_i} - \sum_k \alpha_{ik} \sum_j \beta_{kj} \boldsymbol{x_j}||_2^2 \propto RSS$$

This is because the projection matrix $\boldsymbol{P} = \boldsymbol{V}$ is the orthonormal rotation matrix of $\boldsymbol{X}$ onto its singular vector axes.

Thus, as discussed earlier, using the singular value decomposition permits us to perform AA clustering on a matrix having dimensions of only $N \times N - 1$ instead of $N \times M$. Note that although the learnt parameters of AA, $\boldsymbol{\alpha}$ and $\boldsymbol{\beta}$, do not depend on $M$, the computation times for $\boldsymbol{Z}$ and the $RSS$ do, therefore, working in lower dimensions reduces the computational load.

**Constrained optimization.** Non-negative least squares (NNLS) is a constrained least squares problem in which coefficients are always non-negative (Eq 8). Archetypal Analysis includes an additional constraint coefficient $C$ and adds a row of ones to matrices involved in optimization after every NNLS iteration (Eqs 9 and 10) to ensure the coefficients also sum to one, one of the defining properties of Archetypal Analysis.

Given an $N \times M$ matrix $\boldsymbol{X}$ representing a multivariate data set with $N$ observations and $M$ attributes, for a given $K$, we minimize:

$$\tilde{RSS} = ||\tilde{\boldsymbol{X}} - \tilde{\boldsymbol{\alpha}}\boldsymbol{Z^T}||_F^2 \qquad (6)$$

where $\tilde{\alpha}$ is defined as:

$$\tilde{\boldsymbol{\alpha}} = \begin{pmatrix} \alpha_{11}C & \alpha_{12}C & \alpha_{13}C & \ldots & \alpha_{1K}C \\ \alpha_{21}C & \alpha_{22}C & \alpha_{23}C & \ldots & \alpha_{2K}C \\ \vdots & \vdots & \vdots & \ldots & \vdots \\ \alpha_{N1}C & \alpha_{N2}C & \alpha_{N3}C & \ldots & \alpha_{NK}C \\ 1 & 1 & 1 & \ldots & 1 \end{pmatrix} \qquad (7)$$

and $\boldsymbol{\alpha}$ and archetypes are defined in the previous section. $\tilde{X}$ is defined as:

$$
\tilde{X} = \begin{pmatrix}
x_{11}C & x_{12}C & x_{13}C & \ldots & x_{1M}C \\
x_{21}C & x_{22}C & x_{23}C & \ldots & x_{2M}C \\
\vdots & \vdots & \vdots & \ldots & \vdots \\
x_{N1}C & x_{N2}C & x_{N3}C & \ldots & x_{NM}C \\
1 & 1 & 1 & \ldots & 1
\end{pmatrix}
\tag{8}
$$

where $C$ is a constraint coefficient for $C > 0$ and rows of ones are added after every NNLS iteration. This ensures the constraint $\sum_{j=1}^{k} \alpha_{ij} = 1$ where the value of $C$ represents a weighting between the importance of the constraint and NNLS minimization, with lower $C$'s giving a stronger importance to the constraint. The same method is applied to $\boldsymbol{\beta}$ coefficients to ensure $\sum_{i=1}^{n} \beta_{ij} = 1$.

**Archetype initialization.** We make use of the implementation in [9] which supports three different archetype initialization strategies: (1) random initialization of the archetypes where each dimension of the archetype is sampled from a uniform distribution scaled to have the same range as the input data, (2) random selection of a sample from the input data as the archetype, and (3) the FurthestSum introduced in [11]. By default we make use of FurthestSum initialization as it efficiently generates initial archetype candidates by, after selecting the first archetype randomly, selecting each subsequent archetype as the sample that has the largest aggregate distance from the previously selected archetypes.

**Implementation details.** Archetypal analysis was run with the following parameters (with code adapted from [9]).

- Tolerance: defines when to stop optimization when alternating between finding the best $\boldsymbol{\alpha}$'s for given archetypes $Z$ and finding the best $Z$ for given $\boldsymbol{\alpha}$'s. Specifically, the threshold applied is,

$$
\frac{||RSS_c - RSS_p||_2^2}{RSS_p} > T
\tag{9}
$$

where $RSS$ is the residual sum of squares defined in (Eq 2) for the current iteration $RSS_c$ and the previous iteration $RSS_p$, and $T$ is the desired tolerance. We use a value of $T = 0.001$.

- Maximum number of iterations for the residual sum of squares ($RSS$) minimization: 50.

- Constraint coefficient $C$: coefficient that ensures the summation of $\boldsymbol{\alpha}$'s and $\boldsymbol{\beta}$'s equals one. See Appendix B for further details on the constrained optimization method. We use a value of $C = 0.001$.

- Initialization method: we use FurthestSum [11] as the initialization method.

## Datasets

**Human.** Whole genomes from the Human Genome Diversity Project [12], the Simons Genome Diversity Project [13] and the 1000 Genomes Project [14] have been included in this study. The Human Genome Diversity Project whole genome cohort includes 929 individuals from 54 human populations. The Simons Genome Diversity Project contains 300 publicly

available genomes from 142 diverse populations, and the 1000 Genomes Project includes 2504 individuals from 26 populations. The three datasets were merged, removing duplicated individuals between the studies and retaining only SNPs present in all three datasets, to yield an intersection of 1, 411, 471 SNPs for analysis. Rare variants, having minor allele frequencies < 0.1, were removed. In total, 3558 individuals were included in our study from 7 different continents: 683 from Europe, 805 from Africa, 34 from Oceania, 695 from South Asia, 772 from East Asia, 150 from West Asia, and 419 from the Americas.

**Dogs.** The heterogeneous data set of dog breeds from [15] consists of 1355 individual dogs representing 166 dog breeds. Each sequence has a total of 150, 131 SNPs. Populations with vastly different histories are included, originating from all continents except Antarctica [15]. See Tables B and C in S1 Text for additional domestic dog breed details.

## Results

### Human datasets

**Principal components and Archetypal Analysis.** We first compute the principal components of the human data set and display the first two components in a plot coloured by continental population (Fig 2a). The African populations display the highest genetic variability, extending across the first principal component axis (11% explained variance). We then use all principal components, that is the projection of the samples onto all the left singular vectors of the SVD, yielding a total of $N - 1$ dimensions, as input to Archetypal Analysis and plot the proportional membership of each cluster for each individual in a compositional plot (Fig 2b). The African populations are represented by three archetypes (A1, A2 and A8), while the East Asian and South Asian populations have one archetype each (A3 and A5 respectively). Note that Archetypal Analysis captures the high variation within African groups by utilizing multiple archetypes. The European and West Asian populations share a single archetype (A4), while the Oceanian populations are found on the gradient between the East Asian and South Asian archetypes. Finally, the American populations are represented by two archetypes (A6 and A7) and have a gradient running to the European/West Asian archetype as a result of colonial admixture. Example populations found along this gradient are the Puerto Ricans in Puerto Rico and Colombians in Medellín (Colombia).

**Comparison of ancestry estimates.** To compare the ancestry estimates derived from ADMIXTURE and Archetypal Analysis, we display the proportional ancestry cluster assignments, the $Q$ and $\alpha$ matrices respectively, in a bar plot for $K = 8$ cluster (Fig 3b). Each vertical bar represents an individual and the shaded colors denote the cluster proportions. We also display individuals on a three-dimensional PCA plot with projected archetypes ($Z$) and ADMIXTURE cluster centers ($F$) (Fig 3a). A theoretical comparison of both methods can be found in the Discussion section. The linear correlation between the results displayed in Fig 3 is $\approx 0.84$, while the average pairwise similarity computed using a variant of the Jaccard Index (described by [16]) is $\approx 0.86$.

*Archetypal Analysis*: European (red), South Asian (turquoise), and East Asian populations (yellow) are predominantly represented by a single archetype. American populations show a combination of three archetypes, two of which are mostly specific to this population (light green and dark green) and a third, representing colonial admixture, which is European (red). Individuals from Puerto Rico, population 23, and Colombia (Medellín), population 24, mostly share the third archetype with Europeans. The African populations are represented by three archetypes. One archetype encompasses West African populations such as Mandeka, population 1, Gambian in Western Division (Mandika), population 2, and Mende in Sierra Leone, population 3 (ocean blue). Another includes eastern and southern groups such as Luhya in

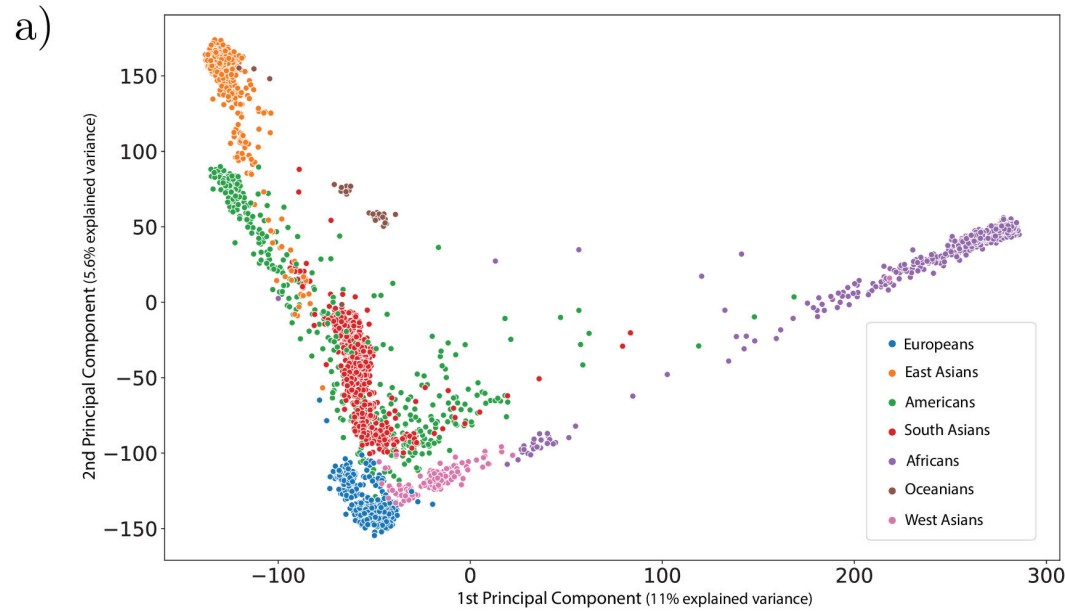

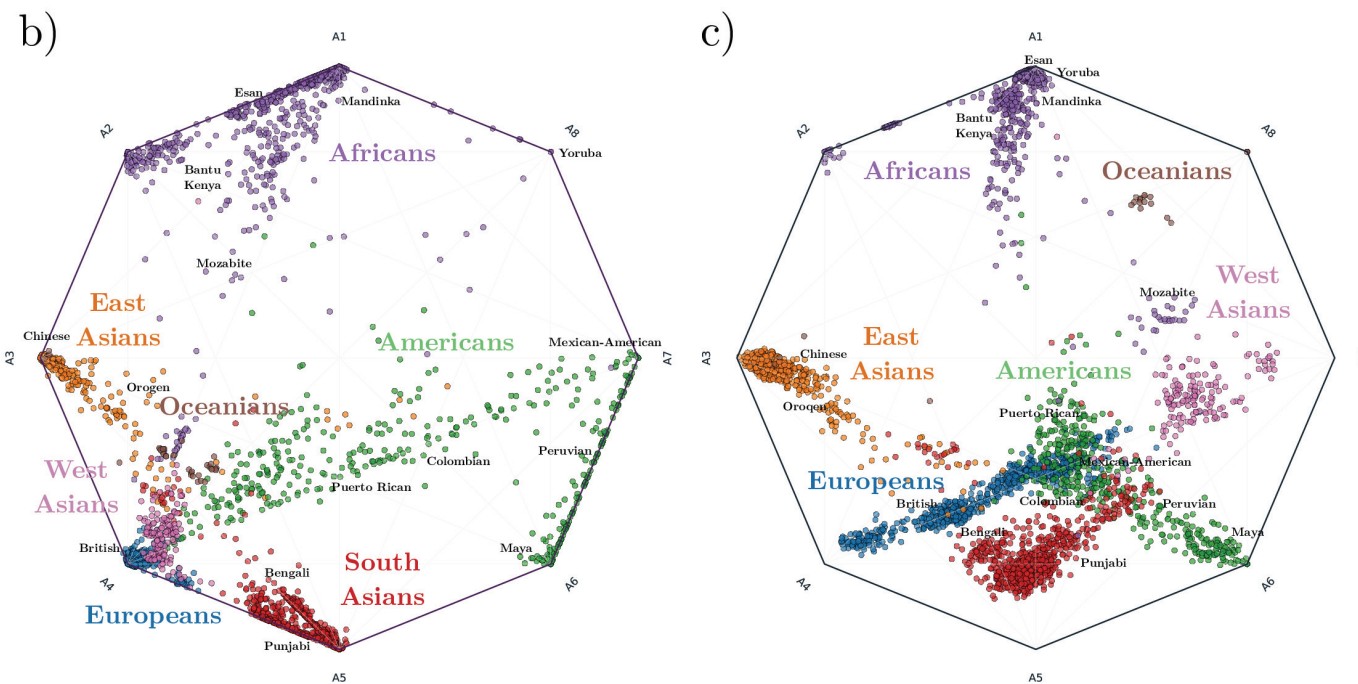

**Fig 2. Principal component analysis and Archetypal Analysis compositional plots for human populations (K = 8). a)**, 2-dimensional PCA plot of human continental populations, where groups of individuals are colored by the unique regional genetic components they possess (see legend) **b)**, Compositional plot giving proportional archetype assignment for each individual (points). Points are coloured by the presence of regional genetic components (colored text) and a few example sub-populations are labeled in small black text. Clusters of individuals from the same population are observed on the vertices of the polygon while diagonals (and edges) between vertices indicate admixed individuals. For details on how to interpret compositional plots see Fig G in S1 Text. **c)**, Similar compositional plot showing the results for ADMIXTURE. Note that several ADMIXTURE clusters (A4, A5, A7) are never attained by real samples. See Figs A and B in S1 Text for additional examples of Archetypal Analysis compositional plots for human continental populations.

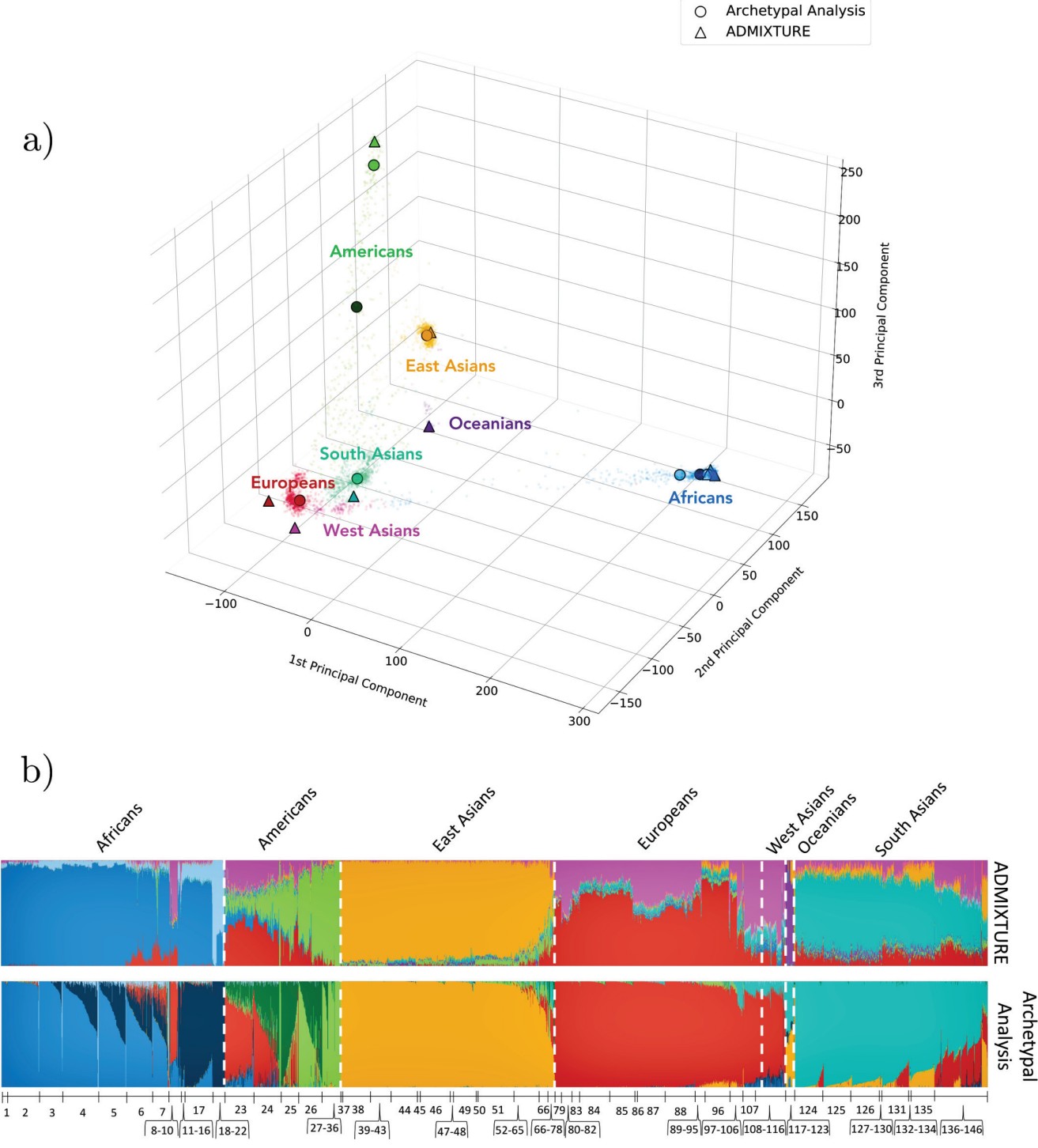

**Fig 3. Comparison of ancestry estimates for human populations (K = 8). a)**, three-dimensional PCA plot of individuals (small points) with projected archetypes (circles) and ADMIXTURE cluster centers (triangles). **b)**, bar plot where individuals are represented along the horizontal axis as narrow bars ordered by population group. The height of the color for each bar shows the proportional colored cluster assignment for that individual sample. We compare the cluster assignments of ADMIXTURE (top) and Archetypal Analysis (bottom). Correspondence of numbers to labels can be found in Tables A and B in S1 Text.

Webuye (Kenya), population 17, and San, population 22 (navy blue). A third archetype represents a few individuals from all African populations (light blue).

*ADMIXTURE*: Oceanian (purple) and East Asian populations (yellow) are predominantly represented by a single cluster center. Europeans and West Asians are represented as a combination of two centers (red and pink) that are located outside the point cloud of individuals; this differs from AA, which captures both with a single cluster inside the point cloud. Americans (largely from Latin America) show traces of the European and West Asian cluster components, but are mostly represented by their own unique, here single, cluster (light green), deriving from the original peoples of the Americas. African populations are predominantly represented by two unique clusters (ocean blue and light blue), while a few populations, such as the North African Mozabites, population 9, show traces of European and West Asian components. Finally, South Asians predominantly show a single cluster (turquoise), but also show traces shared with the European and West Asian clusters.

Overall, Archetypal Analysis provides estimates that qualitatively match ethnolinguistic and geographical labels. Additionally, AA properly captures the wide variation within African populations, assigning more than one cluster to this diverse continent; however, this comes at the cost (due to the fixed number of clusters) of lacking a further unique cluster for Oceanians. Due to its stronger constraints than ADMIXTURE, AA also obtains cluster centroids that could represent real individuals, lying either on or within the set of observations. In contrast ADMIXTURE cluster centers can represent population frequencies that have never existed in the past and also cannot be realized in the present by any combination (admixture) of populations (see Fig 2c). Thus, archetypes can be interpreted as representing actual populations, while ADMIXTURE clusters often cannot.

## Domestic dog breed dataset

**Principal components and Archetypal Analysis.**   We compute the principal components of the dog breed data set and display the first two components in a plot coloured by dog clades (Fig 4a). The Asian Spitz clade shows the highest genetic variability extending across the first principal component axis, including breeds such as Chow Chow, Greenland Sledge Dog and Siberian Husky. The latter is found close to the wolf, while the European Mastiff clade, represented by breeds such as Bull Terrier, Boxer and Bulldog, extends across the second principal component axis. Archetypal Analysis is then computed for $K = 5$ and $K = 15$ with principal components as input (Fig 4b and 4c). For $K = 5$, dog archetypes were found to be the Asian Spitz dogs (A1), the Bulldog-derived dogs (A2), the Terriers (A3), hunting water dogs (A4) and herding dogs (A5). The remaining breeds are displayed as a linear combination of these main archetypes, mostly represented by A5 and A4. This matches the structure shown in PCA, where most of the breeds are clustered in the origin, except the dogs in the Bull Terrier and Husky groups. When increasing the number of archetypes to $K = 15$, individual dog breeds begin clustering around single archetypes, showing the finer scale population structure. New archetypes appear for the Boxer (A3), Irish Wolfhound (A4), Otter Hound (A6), Bullmastiff (A7), Bernese Mountain Dog (A10), Glen of Imaal Terrier (A11), French Bulldog (A12), Boston Terrier (A13), Shetland Sheepdog (A14) and Tibetan Spaniel (A15). The rest of the breeds are mostly found near A14 and A15.

**Performance metric analysis.**   The dog breed dataset was used to benchmark the computation times and clustering quality of both ADMIXTURE and Archetypal Analysis. Running times and explained variances (defined as $EV(Y, \hat{Y}) = 1 - \frac{Var(Y-\hat{Y})}{Var(Y)}$) of ADMIXTURE and Archetypal Analysis are measured for an increasing number of archetypes/clusters $K = 1, \ldots, 22$ and $K = 1, \ldots, 30$ respectively. The initialization was set to *random* for both methods to

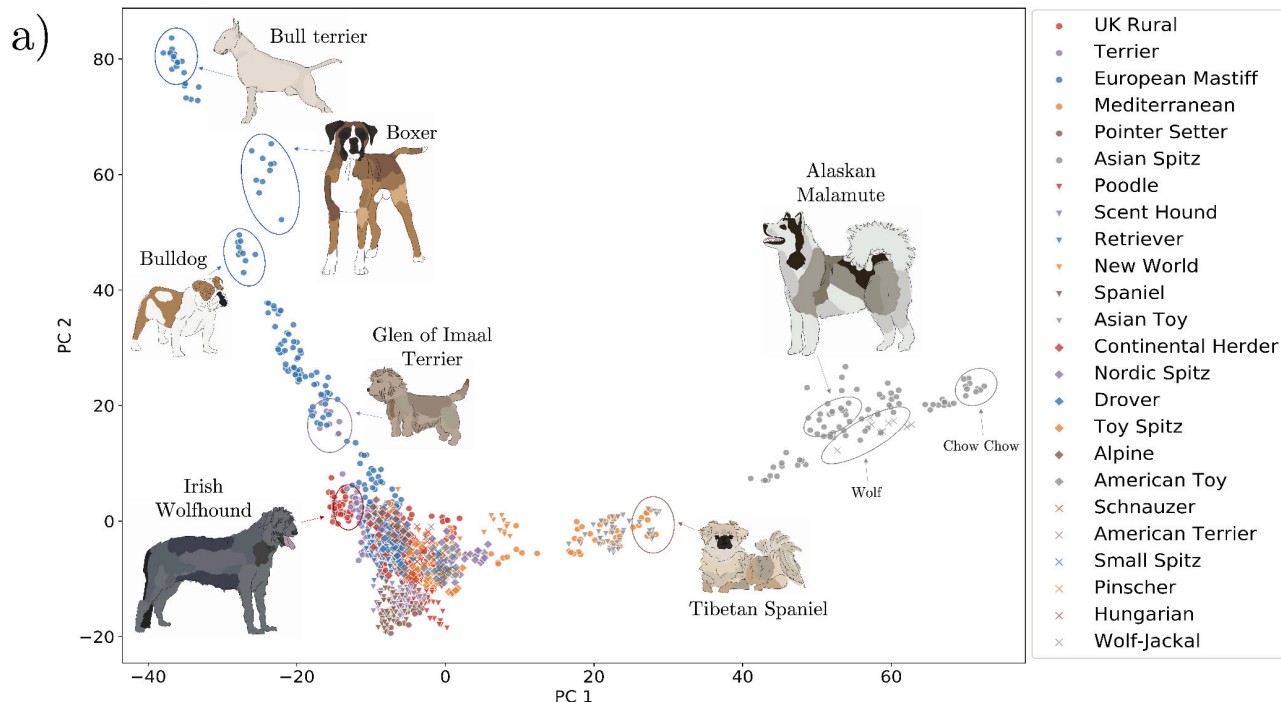

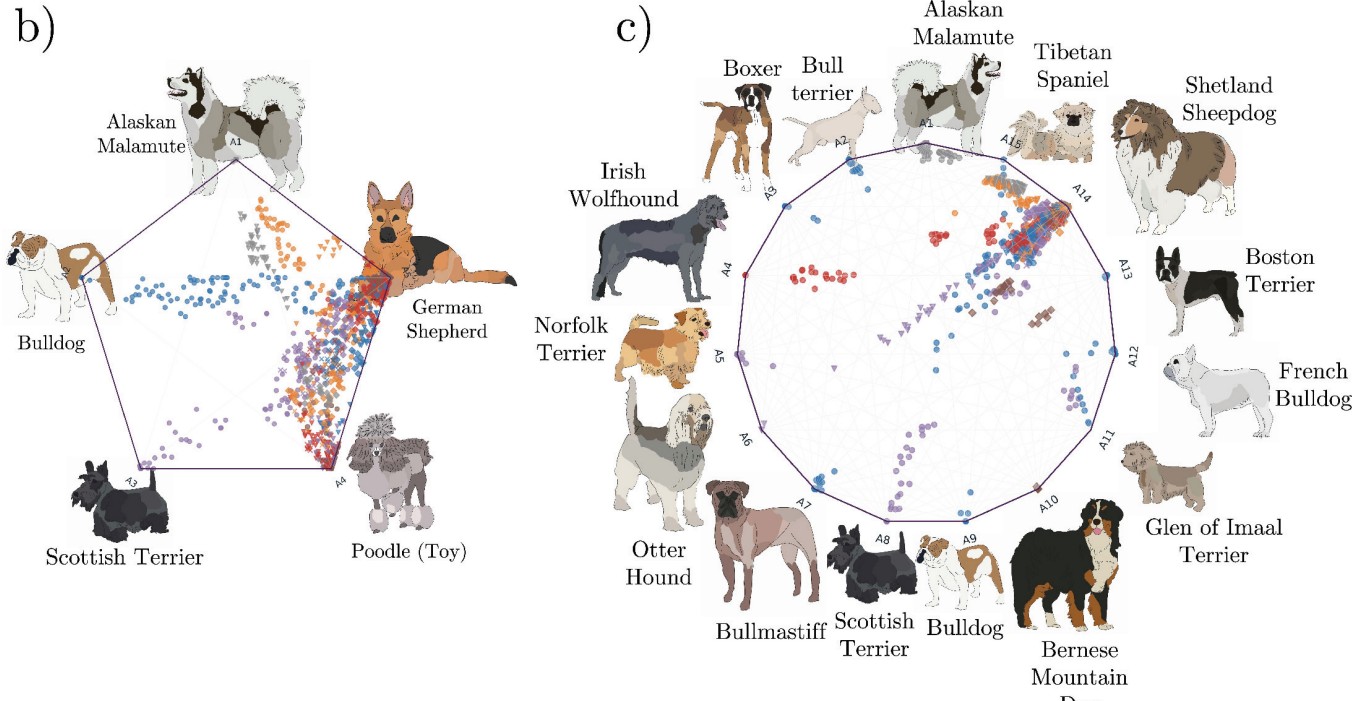

**Fig 4. Principal component analysis and Archetypal Analysis compositional plots for domestic dog breeds. a)**, two-dimensional PCA plot of domestic dog breeds where groups of dogs are colored by clade. **b)** and **c)**, proportional composition of each cluster for each individual in coordinate space for K = 5 and K = 15 archetypes respectively. Data points are coloured by clade and archetype representatives are shown as drawings. Gradients between vertices indicate combinations between breeds. (We thank Ines de Vilallonga for her dog breed illustrations).

achieve uniform comparison and results were averaged over 5 runs. Accumulated run-times increased exponentially with $K$ for ADMIXTURE whereas they increased linearly for Archetypal Analysis (Fig 5). An accumulated runtime of 34 minutes was taken by Archetypal Analysis to compute ancestry estimates for $K = 2$ to $K = 30$ clusters. For ADMIXTURE, the accumulated runtime from $K = 2$ to $K = 30$ was 78 hours. Thus, Archetypal Analysis ran 137 times faster than ADMIXTURE on the domestic dog breed dataset. A similar increase in relative speed was maintained, on average, for non-cumulative times (Table 1). A comparison of the evolution of the runtime of ADMIXTURE and Archetypal Analysis when increasing the number of samples can be found in Fig C in S1 Text. An additional qualitative comparison of different runs of Archetypal Analysis ($K = 15$) on the dogs dataset using different initialization methods can be found in Fig D in S1 Text.

Explained variances increased linearly in the number of clusters for both algorithms (Fig 5). The explained variance for Archetypal Analysis was on average 2% lower than for ADMIXTURE. For the values of $K$ included in this analysis, the mean standard deviation for five averaged runs with random initialization was 0.007 for Archetypal Analysis and 0.0004 for ADMIXTURE. As described in the following Discussion section, the difference in explained variance is due, at least in part, to the stronger restrictions that Archetypal Analysis imposes when estimating the cluster centroids. However, as shown with human sequences in Fig 3, the stronger restrictions of AA lead to a benefit: centroids that are always a linear combination of actual samples, guaranteeing that they represent theoretically observable population samples.

## Discussion

### Population structure overview

Archetypal Analysis was able to capture the high genetic variability in African populations by identifying three ancestral clusters in this large and diverse super-population, compared to only two clusters assigned by ADMIXTURE. This had an effect on the clustering of the proportionally over-represented European populations, mostly collected under a single ancestral cluster in Archetypal Analysis but given two clusters in ADMIXTURE for the same $K$. This illustrates how Archetypal Analysis is more robust to sample scheme bias when properly capturing genetic variation.

Archetypal Analysis proved to be an interpretable alternative to ADMIXTURE. It assigned separate regional archetypes that associated predominantly with Europeans, with South Asians, and with East Asians, and it recognized the high genetic variability of African populations. Differences within regions were also detectable (Fig 3). For example, Indigenous Americans were separated from the remainder of the modern American communities as the light green archetype (e.g. Maya, population 27, Zapotec, population 29, Quechua, population 30). Peruvians in Lima Peru, population 26, were also included in this group, most likely because indigenous groups make up 45% of the Peruvian population. Similarities in related peoples that were geographically spread were also detected. For example, the Bantu peoples (Bantu Herrero, population 8, Bantu South Africa, population 12, Bantu Kenya, population 16, Luhya in Webuye (Kenya), population 17, Bantu Tswana, population 18) comprise several hundred indigenous ethnic groups in Africa spread over a vast area from Central Africa to Southern Africa, nevertheless those present in our dataset were grouped together forming the dark blue archetype.

The red archetype, which is modal in Europeans, is also seen in North African peoples (Saharawi, population 10, and Mozabites, population 9) at a smaller fraction due to geographic proximity and migration. As also observed in previous studies, American populations, such as Puerto Ricans in Puerto Rico, population 23, and Colombians in Medellín (Colombia),

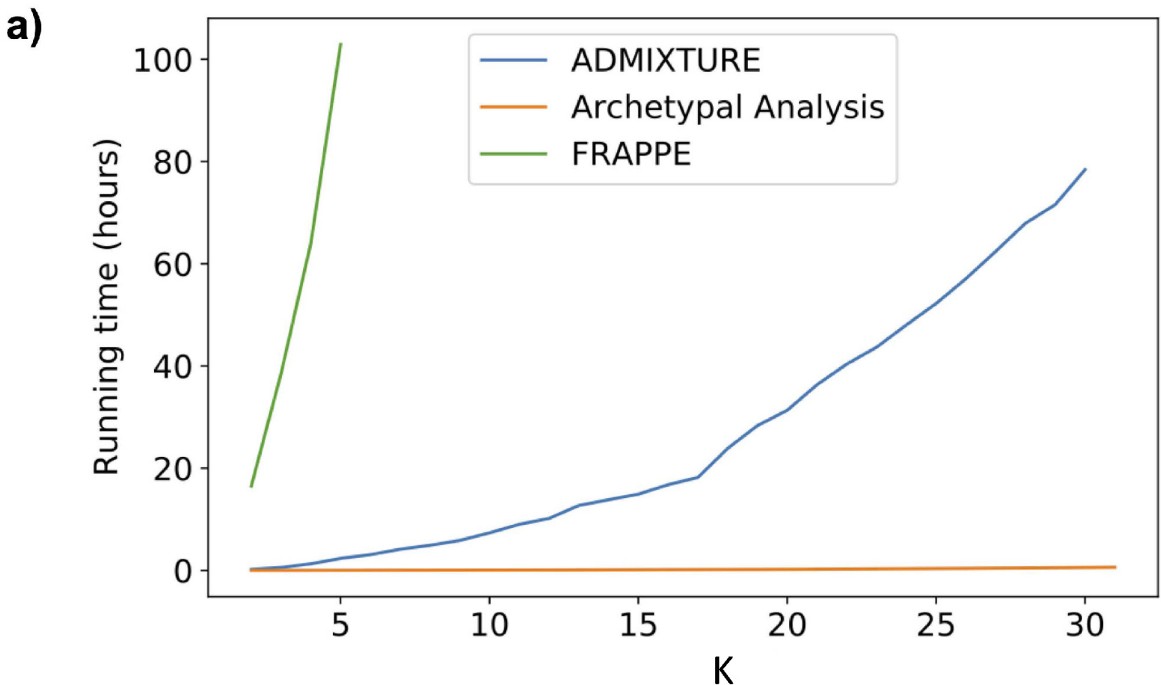

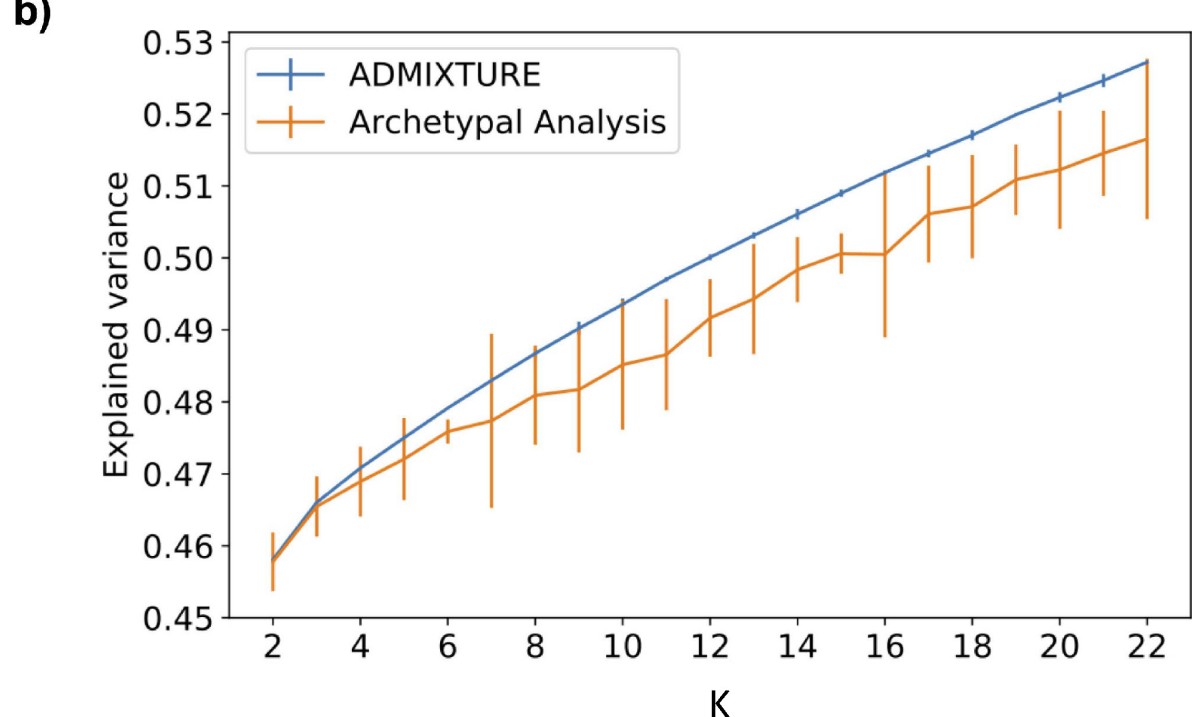

**Fig 5. Performance metrics analysis. a)**, runtime analysis for FRAPPE, ADMIXTURE and Archetypal Analysis for $K = 2$ to $K = 30$. Time is expressed in units of accumulated hours. Note that for FRAPPE we only include up to $K = 5$ due to computational limitations. **b)**, explained variance analysis comparison for ADMIXTURE and Archetypal analysis for $K = 2$ to $K = 22$. Results are averaged over five distinct random seed values for each value of K and the ranges observed are shown as vertical bars.

**Table 1. Runtime (in minutes) for ADMIXTURE-AA comparison.**

| Algorithm | K (number of clusters / archetypes) | | | | | | |
|---|---|---|---|---|---|---|---|
| | [2–6) | [6–10) | [10–14) | [14–18) | [18–22) | [22–26) | [26–30) |
| ADMIXTURE | 43 | 64 | 97 | 150 | 247 | 250 | 319 |
| AA | 0.5 | 0.48 | 0.7 | 1 | 1.4 | 1.9 | 2.4 |
| Relative speed | 86× | 133× | 139× | 150× | 176× | 132× | 132× |

population 24, showed a European associated cluster due to Spanish colonization. The effects of this historical event can also be observed in (Fig 2b), which shows a diagonal of decreasing European admixture that runs from the European vertex (A4) to Puerto Ricans in Puerto Rico, through Colombians in Medellín (Colombia), to some of the Mexican Ancestry individuals in Los Angeles CA USA, Peruvians in Lima Peru and Mayans. Some of these Latin American samples (green points) also fall above this diagonal cline, due to colonial-era admixture with West African populations (vertex A1 at top). Archetypal Analysis also identified a large shared component with Europeans in South Asians that ADMIXTURE did not detect (Fig 3b). For example, the Brahui, population 141, Kalash, population 138, and Baloch, population 140, were identified with the red archetype, modal in Europeans, by Archetypal Analysis, but not by ADMIXTURE. This shared component has been noted using other methods for detecting shared ancestry and has been associated with the Ancestral North Indians [17], an ancestral genetic grouping that shares some ancestry with other Indo-European speakers from India to Iran to Europe.

## Relationship between Archetypal Analysis and ADMIXTURE

The popular algorithm ADMIXTURE estimates individual ancestries by computing maximum likelihood estimates in a parametric model. Specifically, it maximizes the biconcave log-likelihood of the model using block relaxation:

$$\mathcal{L}(\mathbf{Q}, \mathbf{F}) = \sum_{i,j}(n_{ij} \ln p_{ij} + (2 - n_{ij}) \ln(1 - p_{ij})) \tag{10}$$

where genotype $n_{ij}$ for individual $i$ at SNP $j$ represents the number of type '1' alleles observed. Given $K$ populations, the success probability $p_{ij} = \sum_{k=1}^{K} q_{ik} f_{ki}$ in the binomial distribution $n_{ij} \sim$ Bin$(2, p_{ij})$ depends on the fraction $q_{ik}$ of $i$'s ancestry attributable to population $k$ and on the frequency $f_{kj}$ of the allele 1 in population $k$, where $q_{ik}$ and $f_{kj}$ are the entries of $\mathbf{Q}$ and $\mathbf{F}$ respectively [3].

ADMIXTURE and Archetypal Analysis share similar modeling assumptions. Both $\mathbf{Q}_{kj}$ ADMIXTURE and $\boldsymbol{\alpha}$ archetype fractions can be interpreted as partial cluster assignments, while ADMIXTURE frequency coefficients $\mathbf{F}_{kj}$ and archetype coordinates $\mathbf{Z}$ encode cluster center locations in SNP space. A key difference is that ADMIXTURE cluster centroids have M (# of SNPs) free parameters, in other words, the frequency at each SNP for each cluster ($f_{kj}$) is a parameter that needs to be learnt. Instead, in AA, cluster centroids have N (number of samples) free parameters, that is, a coefficient ($\boldsymbol{\beta}$) for each training sample needs to be learnt for each cluster center. When $M \gg N$ (the typical scenario when working with genomic data), AA has far fewer free-parameters than ADMIXTURE. This can lead to lower explained variance values (or higher reconstruction errors), but guarantees centers that exist within the convex hull of real samples (and thus could represent a real descendant individual), while ADMIXTURE can over-fit, yielding centers outside the hull of the observed data (see Results section)

that may represent no population that has ever existed. Furthermore, because AA does not optimize each of the $M$ free-parameters, it can work with rotated data (the left singular vectors of the SVD) without any loss of information, or with dimensionally-reduced (projected) data, allowing for a much more efficient computation.

The likelihood function of ADMIXTURE can be understood as an error or distance metric between the input sequences $X$ (where both haplotypes have been averaged) and a decomposed product $QF$. In fact, when $X \approx QF$:

$$\frac{1}{2}\mathcal{L}(Q, F) = \sum_{i,j}(x_{ij} \ln q_{ij}f_{ij} + (1 - x_{ij}) \ln (1 - q_{ij}f_{ij})) \approx \sum_{i,j}||x_{ij} - q_{ij}f_{ij}||_2^2 \qquad (11)$$

Therefore, the likelihood function resembles the *RSS* problem of AA. In fact, ADMIXTURE can be understood as a type of likelihood-based relaxed archetypal analysis, where the constraints imposed on the cluster centroids are loosened.

Another shared aspect of both methods is the alternating nature of the optimization procedure. In both methods, cluster centers and cluster assignments are optimized in an iterative manner. Once the cluster assignments are fixed, optimizing centers becomes a convex problem, and vice versa, allowing for fast convergences. A summary of this comparison can be found in Table 2.

## Relationship between Archetypal Analysis, ADMIXTURE, K-Means, and K-Medioid clustering

Archetypal Analysis and ADMIXTURE hold a strong relationship with K-Means and K-Medioids. As already stated in [11], if the constraints on the archetypes $Z$ are relaxed, and cluster assignments are limited to binary values $\alpha_{ij} \in \{0, 1\}$ and $\sum_{j=1}^{k} \alpha_{ij} = 1$, then archetypal analysis becomes equivalent to K-Means. Similarly, if the sparsity regularization used in ADMIXTURE [3] is strongly applied, the cluster assignments $Q$ become binary and the technique becomes similar to K-Means. In a similar fashion, if both $\alpha$ and $\beta$ are restricted to be binary, $\alpha_{ij}, \beta_{ij} \in \{0, 1\}$, Archetypal Analysis becomes equivalent to K-Medioids. Therefore, AA can be understood as a smooth or fuzzy version of K-Medioids. Note that both K-Means and K-Medioids are also typically optimized in a iterative alternating nature, similar to AA and ADMIXTURE.

Fig 6 shows a qualitative comparison of all four of these methods when $K = 4$. Examples with $K = 3$ and $K = 5$ can be found in Figs E and F in S1 Text. We can observe that

**Table 2. ADMIXTURE and Archetypal Analysis comparison.**

|  | ADMIXTURE | Archetypal Analysis |
|---|---|---|
| **Model** | $X \approx QF$ | $X \approx \alpha Z^T$ |
| Loss Function | log-likelihood | RSS |
| Free-parameters | $(N + M)K - N$ | $2NK - N - K$ |
| **Cluster Assignments (CA)** | $Q$ | $\alpha$ |
| CA Dimensions | $N \times K$ | $N \times K$ |
| CA Free-parameters | $N(K - 1)$ | $N(K - 1)$ |
| CA Constraints | $\sum_{j=1}^{K} Q_{ij} = 1$ and $Q_{ij} \geq 0$ | $\sum_{j=1}^{K} \alpha_{ij} = 1$ and $\alpha_{ij} \geq 0$ |
| **Cluster Centroids (CC)** | $F$ | $Z = X^T\beta$ |
| CC Dimensions | $K \times M$ | $K \times M$ |
| CC Free-parameters | $KM$ | $K(N - 1)$ |
| CC Constraints | $0 \leq F_{ij} \leq 1$ | $\sum_{j=1}^{N} \beta_{ij} = 1$ and $\beta_{ij} \geq 0$ |

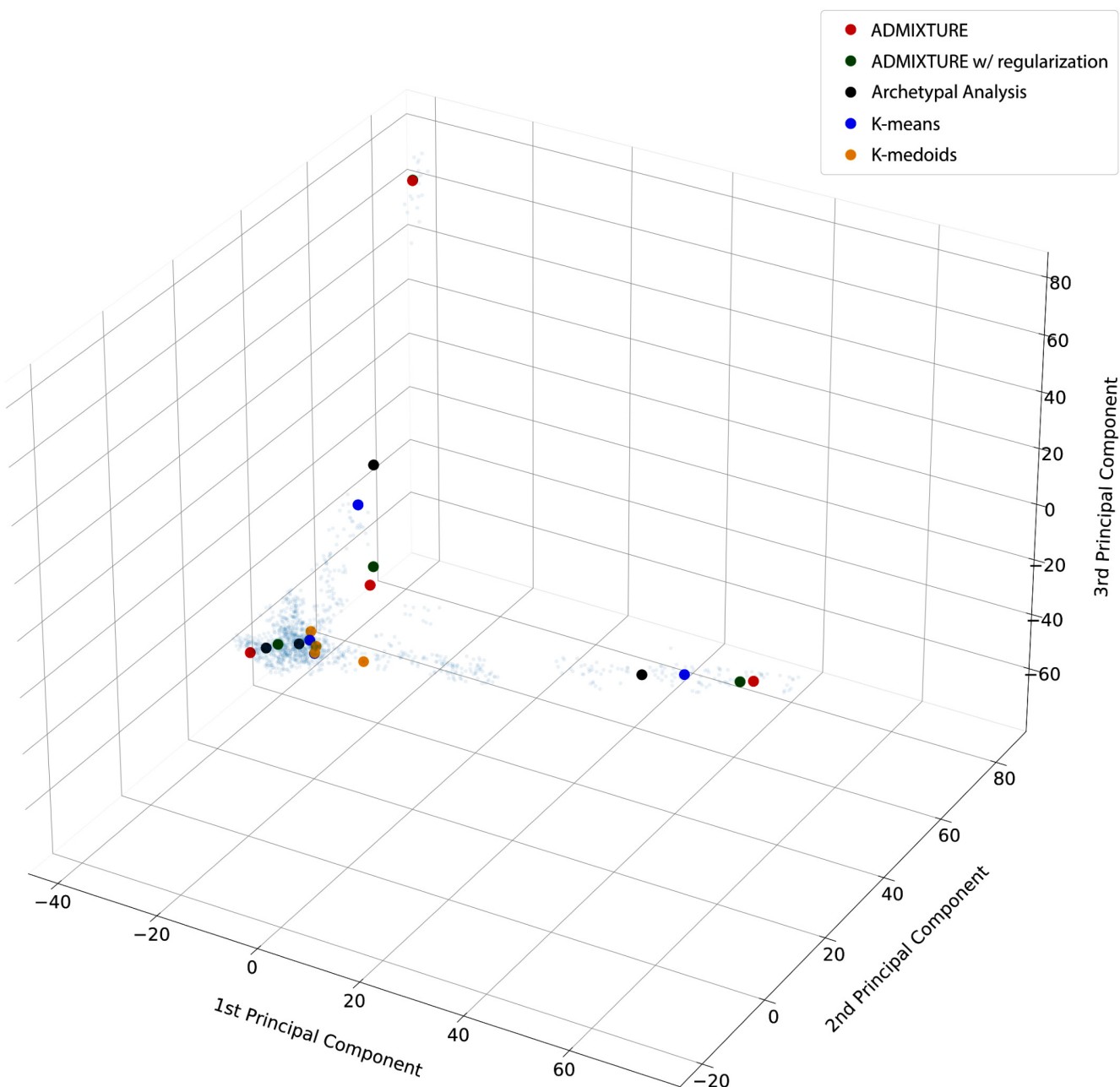

**Fig 6. Comparison of cluster centroids from different methods.** Cluster centers learned by ADMIXTURE, ADMIXTURE with sparsity regularization, Archetypal Analysis, K-Means, and K-Medoids for K = 4 are plotted as solid circles while the underlying samples are plotted as small blue points. Regularization in ADMIXTURE is introduced with lambda = 500 and epsilon = 0.1.

ADMIXTURE with sparsity regularization (green) obtains cluster centroids less extremal than ADMIXTURE without, showing a behaviour that tends to K-Means. Note that the differences between cluster centers will not depend only on differences in modelling assumptions for each technique, but also in differences in implementation details and initialization approaches of each method.

## Conclusion

In this paper we show how Archetypal Analysis (AA) can be used as a fast alternative to ADMIXTURE for population clustering. We also show that the Archetypal Analysis model has fewer degrees of freedom, constraining the centroids of clusters to be within the convex hull of the training samples, leading to lower explained variance than ADMIXTURE, but providing more interpretable cluster centroids that represent realizable populations. We apply our proposed system to human and dog genotypes, showing that AA can perform more than two orders of magnitude faster than ADMIXTURE while still properly capturing the population structure of the data.

## Supporting information

**S1 Text. Supporting information.** Detailed information and additional experiments are provided.
(PDF)

## Acknowledgments

We would like to thank Inés de Vilallonga for her dog breed illustrations.

## Author Contributions

**Conceptualization:** Daniel Mas Montserrat, Alexander G. Ioannidis.

**Data curation:** Julia Gimbernat-Mayol.

**Formal analysis:** Julia Gimbernat-Mayol, Albert Dominguez Mantes, Daniel Mas Montserrat, Alexander G. Ioannidis.

**Funding acquisition:** Carlos D. Bustamante, Alexander G. Ioannidis.

**Investigation:** Julia Gimbernat-Mayol, Albert Dominguez Mantes, Daniel Mas Montserrat, Alexander G. Ioannidis.

**Methodology:** Julia Gimbernat-Mayol, Albert Dominguez Mantes, Daniel Mas Montserrat, Alexander G. Ioannidis.

**Project administration:** Daniel Mas Montserrat, Alexander G. Ioannidis.

**Resources:** Daniel Mas Montserrat.

**Software:** Julia Gimbernat-Mayol, Albert Dominguez Mantes.

**Supervision:** Daniel Mas Montserrat, Alexander G. Ioannidis.

**Validation:** Albert Dominguez Mantes, Daniel Mas Montserrat, Alexander G. Ioannidis.

**Visualization:** Julia Gimbernat-Mayol, Albert Dominguez Mantes, Alexander G. Ioannidis.

**Writing – original draft:** Julia Gimbernat-Mayol, Daniel Mas Montserrat, Alexander G. Ioannidis.

**Writing – review & editing:** Julia Gimbernat-Mayol, Albert Dominguez Mantes, Daniel Mas Montserrat, Alexander G. Ioannidis.

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
