## [Decision Letter · Decision Letter 0]

14 Feb 2022

Dear Dr. Ioannidis,

Thank you very much for submitting your manuscript "Archetypal Analysis for Population Genetics" for consideration at PLOS Computational Biology.

As with all papers reviewed by the journal, your manuscript was reviewed by members of the editorial board and by several independent reviewers. In light of the reviews (below this email), we would like to invite the resubmission of a significantly-revised version that takes into account the reviewers' comments.

We cannot make any decision about publication until we have seen the revised manuscript and your response to the reviewers' comments. Your revised manuscript is also likely to be sent to reviewers for further evaluation.

Sincerely,

Heather E. Wheeler, Ph.D.

Guest Editor

PLOS Computational Biology

Ilya Ioshikhes

Deputy Editor

PLOS Computational Biology

Reviewer's Responses to Questions

**Comments to the Authors:**

Reviewer #1: In this manuscript, Gimbernat-Mayol et al. propose a method for determining the ancestry composition of cohort data by using Archetypal Analysis (AA). They argue that using AA is more computationally efficient than traditionally used methods such as ADMIXTURE, and that it results in similar genetic cluster assignments. Their computational efficiency improvement is impressive, but I think an expansion of the analyses presented in manuscript is necessary to provide the reader with a more full perspective on the benefits and drawbacks of the methodology proposed. Expansion of the discussion as it pertains to the biological analyses would also be helpful. I detail these and other specific questions below.

Broader Comments:

1. The authors note that several programs represent competing methods that could be used instead of AA for the purposes of ancestry clustering. However in the manuscript only ADMIXTURE is benchmarked. It would be useful to see comparisons to the other two methods listed (line 8) – STRUCTURE and FRAPPE – to assist in justification of the improvement of AA compared to other existing tools. At least efficiency benchmarking would be helpful if the authors do not wish to present full results from the empirical datasets.

2. How are users to determine the best fit value of k in their data with AA? With ADMIXTURE, users can run a cross-validation procedure to determine the value of k that has the best predictive accuracy. These CV errors are typically plotted across increasing ks to visualize the elbow in the dataset. Such a plot would also be a helpful additional figure to show the concordance between AA and ADMIXTURE in terms of the best fit k to the data.

3. The authors note in their introductory discussion of PCA that “interpretation can often be misleading if sampling designs are irregular” (line 26). However no discussion of the impact of sample composition for ADMIXTURE or AA is presented in the manuscript. Both will be impacted to at least some degree by sample composition, and this should at least be described for downstream users in clear terms in the manuscript, if not explored empirically. For example, sample size imbalances will impact the ordering of pulling out genetic clusters with increasing values of k. Additionally, both methods will be affected by the inclusion of related individuals in the analysis. Discussion/exploration of the impact of sample composition and recommendations for QC/data inputs by downstream users is necessary to ensure proper utilization of the method.

4. Some of the phrasing in the text related to the human populations should be revisited, particularly in sections prior to the Discussion.

a. Specifically, the populations of the Americas are referred to throughout interchangeably as “Native American populations” (Figure 3) or “indigenous individuals from the Americas” (line 144) but include many populations and individuals who would not self-identify as such, but may be better described as ‘Latinx’, ‘Latin American’, or simply ‘American.’ While there is a Native American ancestry component present in these populations of the Americas, it is incorrect to refer to many of these groups themselves as ‘Native American’ populations. The authors should revisit the phrasing surrounding such groups to be sensitive to this distinction.

b. In a similar vein, the authors refer to individuals from distinct areas of the African continent as being part of the same population at several points in the text – i.e. “The African population displays the highest genetic variability...” (line 154). There are of course many different populations within Africa with substantial genetic and ethnolinguistic diversity across them which this phrasing currently glosses over. Referring to continental groupings instead as ‘super-population’ or even simply making the population plural ( “The African super-population displays” or “The African populations display…”) when discussing them would ensure this comes through.

5. The Discussion currently is very thin on biological takeaways from the two empirical datasets. This would be a good place to expand on both the interpretation of the differences seen between AA and ADMIXTURE, and what can be inferred based on the patterns you observe. Cite previous research to justify that the AA determinations are logical based on what we know about the population structure and history, particularly if they differ from the determinations by ADMIXTURE.

Other Specific Comments:

6. I did not receive any Supplementary Information. Please ensure supplementary information is included in the revision.

7. Author Summary – you note that AA has ‘representational advantages’ over ADMIXTURE. What is meant by this? This should be described in the main text if this is perceived as a primary benefit of AA over ADMIXTURE.

8. It appears that only bi-allelic variants can be included in the AA analysis. Is this the case? Can indels be used or just SNPs? Discussion of them specific dataset filtering requirements would be useful.

9. Line 141 – you used a MAF cutoff of 10% here. This is extremely high. Would results be the same had you included slightly less common variants in the analysis? Justify threshold choice.

10. Line 155 – you note that “all principal components” were used but do not note how many you computed. Include the number of PCs.

11. Figure 3 presentation – I have several suggestions to aid in the parsing of this figure.

a. The color scheme is not consistent across panels, which would aid viewer interpretation.

b. It is currently very difficult in panel B to determine the specific populations. The figure legend should at least point to Table 3, which contains the key to the population numbering system. A potential expansion of the x axis to allow for a more detailed population labeling may also help.

12. Fig 3 interpretation: There seem to be several notable differences in the ancestry component determination between ADMIXTURE and AA.

a. Humans: The text notes the 2 vs 1 components seen in ‘Europeans’ and ‘Native Americans’, but does not discuss which orientation is more reasonable based on prior research into the population structure and history of these areas. Additionally, there’s also a distinction in Africa where ADMIXTURE appears to identify a unique San component (though it is hard to see if it is indeed the San given the population labeling system), and AA picks up a dark blue component present in many African populations that appears to be driven by the Luhya and San. Further discussion of the interpretation of differences from the two methods would be useful to include in the Discussion.

13. Related, in the dog dataset, are the patterns you observe compatible with what is known about the dog phylogeny? That is, do the trends fit with the expectations from their demography?

14. How correlated are the ancestry fractions across AA and ADMIXTURE? A quantitative comparison would complement the qualitative comparison.

15. How did you decide on the best fit value of k for your empirical datasets? Justify your choice of the k chosen to be presented in the primary figures (8 for humans, 15 for dogs).

16. In studies examining admixture it is typical to run multiple ADMIXTURE runs with different seeds to confirm the results are consistent. I would suggest the authors do this, as the ancestry composition determined is a primary focus of their paper. Running 10x at different seeds for each value of k, for example, would show if there are different likely modes in the data or if the same result is always determined. It appears this was done to generate Figure 5, but that Figures 3 and 4 may be based only on one run each.

17. Fig 5 – how does runtime scale with increasing sample size?

18. Line 264 – “a gradient of relatedness to Europeans.” This phrasing is a bit confusing as you don’t mean relatedness in the technical genetic meaning. I would change the wording to something like ‘European ancestry component’ or ‘cline of European admixture’ or similar.

Reviewer #2: Review is uploaded as an attachment

**Have the authors made all data and (if applicable) computational code underlying the findings in their manuscript fully available?**

Reviewer #1: **No: **The authors note their code will be made available on github but it is not currently visible there.

Reviewer #2: Yes

PLOS authors have the option to publish the peer review history of their article (what does this mean?). If published, this will include your full peer review and any attached files.

Reviewer #1: No

Reviewer #2: **Yes: **Alex Diaz-Papkovich
---

## [Decision Letter · Decision Letter 1]

14 Jun 2022

Dear Dr. Ioannidis,

We are pleased to inform you that your manuscript 'Archetypal Analysis for Population Genetics' has been provisionally accepted for publication in PLOS Computational Biology.

Please note that your manuscript will not be scheduled for publication until you have made the required changes, so a swift response is appreciated. Please also consider making the minor additions/changes suggested by the reviewers in your final edit.

Best regards,

Heather E. Wheeler, Ph.D.

Guest Editor

PLOS Computational Biology

Ilya Ioshikhes

Deputy Editor

PLOS Computational Biology

Reviewer's Responses to Questions

**Comments to the Authors:**

Reviewer #1: Summary:

In this manuscript, Gimbernat-Mayol et al. propose a method for determining the ancestry composition of cohort data by using Archetypal Analysis (AA). They argue that using AA is more computationally efficient than traditionally used methods such as ADMIXTURE, and that it results in similar genetic cluster assignments. Their computational efficiency improvement is impressive. Their method is also able to resolve ancestral archetypes that represent truly possible individuals, which they point out as a benefit over competing methods.

In their revisions, the authors have expanded the scope of analyses to address the technical questions brought up by the other review and myself and have revised some of their text. A couple small points remain but the bulk of concerns have been addressed. Specifically:

• The authors now test the question of different initialization modifying results, which both reviewers brought up. There seems to be quite a big effect, which may be worth expanding on even further, but is briefly mention in the discussion.

• Importantly, the authors also now include a discussion of the impact of sample size imbalance on component/archetype definitions, and argue that in fact AA does better at resolving the complex ancestral diversity in Africa as compared to ADMIXTURE in spite of the bias in sample size towards the latter.

• The authors however have not commented on the impact of inclusion of related individuals.

• Runtime scaling across sample sizes now shown also, as well as a benchmark against FRAPPE.

• The authors now also a plot on how to interpret the compositional plots, which I think many readers will find extremely useful, as this is a somewhat non-standard way to visualize ancestry proportions.

• The authors include a formal comparison of the similarity between AA and ADMIXTURE output.

• The authors have appropriately revised their description of cohorts throughout the manuscript.

• A much fuller discussion of the interpretation of the dog ancestry inference results is now also presented, which is interesting and helpful to confirm that the parsing of clusters makes demographic sense.

• The human discussion is also expanded, though still feels a bit meager in comparison to the expanded dog discussion.

• Related, the authors find that using AA “the American populations are represented by two archetypes (A6 and A7) and have a gradient running to the European/West Asian archetype as a result of colonial admixture. Example populations found along this gradient are the Puerto Ricans in Puerto Rico and Colombians in Medellin (Colombia).” Similar to the point brought up by reviewer 1, prior research has demonstrated a west African component in Caribbean populations, which is further supported by historical records of the translatlantic slave trade. The authors do not note this additional component in some American populations in their discussion, which is a point I expect many applied readers would be interested in.

Reviewer #2: Thank you for your hard work on improving the manuscript. I have only minor fixes:

* Line 207: Typo - "the the"

* Line 288: Typo - there is a stray "("

**Have the authors made all data and (if applicable) computational code underlying the findings in their manuscript fully available?**

Reviewer #1: Yes

Reviewer #2: Yes

PLOS authors have the option to publish the peer review history of their article (what does this mean?). If published, this will include your full peer review and any attached files.

Reviewer #1: No

Reviewer #2: **Yes: **Alex Diaz-Papkovich

---

## [Editor Report · Acceptance letter]

18 Aug 2022

PCOMPBIOL-D-21-02146R1 

Archetypal Analysis for Population Genetics

Dear Dr Ioannidis,

I am pleased to inform you that your manuscript has been formally accepted for publication in PLOS Computational Biology. Your manuscript is now with our production department and you will be notified of the publication date in due course.

With kind regards,

Olena Szabo
